# A hybrid neural network for driving behavior risk prediction based on distracted driving behavior data

Xin Fu[1], Hongwei Meng [1]*, Xue Wang[1]*, Hao Yang[2], Jianwei Wang[3]

**1** College of Transportation Engineering, Chang'an University, Xi'an, China, **2** Department of Civil and Environmental Engineering, University of Washington, Seattle, WA, United States of America, **3** Engineering Research Center of Highway Infrastructure Digitalization, Ministry of Education, Chang'an University, Xi'an, China

* 2020134010@chd.edu.cn (HM); wangxue@chd.edu.cn (XW)

**Data Availability Statement:** Data cannot be shared publicly because of relevant data protection laws. Data are available from the Shaanxi Provincial Road Transport Development Center of China for researchers who meet the criteria for access to confidential data. The address of the Shaanxi

## Abstract

Distracted driving behavior is one of the main factors of road accidents. Accurately predicting the risk of driving behavior is of great significance to the active safety of road transportation. The large amount of information collected by the sensors installed on the vehicle can be identified by the algorithm to obtain the distracted driving behavior data, which can be used to predict the driving behavior risk of the vehicle and the area. In this paper, a new neural network named Driving Behavior Risk Prediction Neural Network (DBRPNN) is developed for prediction based on the distracted driving behavior data. The network consists of three modules: the Feature Processing Module, the Memory Module, and the Prediction Module. In this process, attribute data (time in a day, daily driving time, and daily driving mileage) that can reflect external factors and driver statuses, are added to the network to increase the accuracy of the model. We predicted the driving behavior risk of different objects (Vehicle and Area). For the applicability improvement of the model, we further classify the distracted driving behavior categories, and DBRPNN can provide more accurate risk prediction. The results show that compared with traditional models (Classification and Regression Tree, Support Vector Machines, Recurrent Neural Network, and Long Short-Term Memory), DBRPNN has better prediction performance. The method proposed in this paper has been fully verified and may be transplanted into active safety early warning system for more accurate and flexible application.

## Introduction

Driving behavior analysis is an important part of traffic safety research. It reflects the status of drivers and vehicles in the process of vehicle operation. Distracted driving behavior refers to a series of operations conducted by drivers on public roads that may lead to abnormal traffic conditions and thus road accidents [1]. The analysis of driving behavior is helpful to measure drivers' driving safety and prevent traffic accidents. As we all know, there is a close connection between distracted driving behavior and traffic accidents. With the

Provincial Road Transport Development Center of China is No. 18 Yaowangdong, Xi'an 710003, Shaanxi Province, China, and the telephone number is 029-87325981.

**Funding:** The research is supported by the Key R&D Project of the Ministry of Science and Technology of the People's Republic of China (2020YFC1512004); Major Projects of the National Social Science Fund(20&ZD099). The funders play a role in the decision to publish.

**Competing interests:** The authors have declared that no competing interests exist.

advancement of the Internet of Things technology, the data collected by large-scale drivers' driving behaviors gradually become more available. The development trend of the collection of driving behavior data has great significance and influence on the prevention of traffic accidents. Distracted driving is one of the most important factors leading to traffic accidents [2]. Effective prediction of distracted driving behavior of vehicles can timely remind drivers or forcibly take over the vehicle with safety control devices at critical moments, to effectively prevent traffic accidents.

Simulation of the driver's driving behavior is the most direct way to forecast for distracted driving behavior, however, a different driver's driving skills, driving style, emergency ability, mood swings, mental status, education background, life experience, each is not identical [3–7], such as environment such as road conditions, weather, illumination, time of day also make a big difference [8], These uncertain factors make it difficult to simulate individual driving behavior objectively. However, in the process of driving, no matter what factors the vehicle and the driver is affected by, the distracted driving behavior will eventually be reflected by the vehicle and the driver's behavior. Based on this fact, this paper carries on the risk prediction research through the distracted driving behavior data.

Our contribution is mainly located in four aspects:

(1) The Driving Behavior Risk Prediction Neural Network (DBRPNN) is proposed, which consists of three parts: the Feature Processing Module, the Memory Module, and the Prediction Module. The model can predict the driving behavior risk with high precision according to the distracted driving behavior data.

(2) The performance of large, real provincial datasets tested in this neural network is encouraging. For every 30 minutes, the Accuracy is 0.9146 and the Weighted-Precision is 0.9156.

(3) We observed the impact of different time intervals on the prediction results. When the time interval is 30 minutes, the risk prediction Accuracy is the highest, and the Accuracy is 0.9146.

(4) It not only predicts the risk level of the vehicle but also predicts the risk level of the area. It is suggested that the prediction results have a clear supporting role for both drivers and road management.

(5) Test different distracted driving behavior categories: the distracted driving behavior shown by the vehicle and the distracted driving behavior shown by the driver. The results show that DBRPNN is capable of handling the risk prediction tasks of different categories.

## Related work

At present, studies on distracted driving behavior at home and abroad are mainly divided into studies on drivers and cars. N. Kuge collected steering wheel Angle data on the simulator and established the lane change intention model through hidden Markov theory [9]. Andrew Liu proposed that driver control behavior can be predicted through vehicle movement behavior [10]. Omerustaoglu et al. combined in-car data and image data to study distracted driving behaviors through deep learning [11]. Jeong et al. used the data collected by the built-in 3-axis gyroscope of the vehicle to identify two driving behaviors by support vector machine [12]. Other studies described various aggressive driving behaviors and formulated their standards (Tasca [13], Abou-Zeid [14], Li [15], Yang [16]). Chen et al. proposed a graphical modeling method based on onboard GPS and OBD data and modeled individual driving behaviors through statistical methods [17]. Han and Yang collected vehicle speed,

acceleration, and deflection speed through vehicle equipment to identify four dangerous driving states [18].

In terms of risk prediction, behavioral prediction and flow prediction have made abundant research progress. Tang proposed a forecasting framework named the spatiotemporal gated graph attention network to predict the urban traffic flow based on license plate recognition data [19]. In addition, Pu uses historical data to predict road surface friction [20, 21]. Tang used a geographically weighted Poisson quantile regression model to study the spatial heterogeneity and estimated the spatial impact on crash frequency [22].

There are a large number of predictions of distracted driving behaviors and accidents in existing studies, which can be divided into three categories:

(1) Linear theoretical model based on time series model and Kalman filtering model [23, 24].

(2) Nonlinear statistical model based on a nonparametric regression model and chaos theory model [25–28].

(3) Machine learning prediction model based on neural network and support vector machine [26, 29–32].

To sum up, existing studies, especially neural network modeling methods, have made great progress in the prediction of distracted driving behaviors and accidents, but further exploration is still needed in the following aspects:

(1) Most studies only consider vehicles or drivers, and few studies consider both perspectives [30].

(2) Some studies only consider the perspective of time, without combining the perspective of space [23].

(3) Data used in some studies are obtained through simulation experiments [32], but real data are necessary to understand the actual situation of distracted driving behavior.

With the continuous development of neural network technology, neural network models can dig out deeper rules of data. Neural Network has great advantages in dealing with traffic flow prediction [33] and traffic accident prediction [34]. Among many Neural Network models, Recurrent Neural Network (RNN) can simulate continuous information by maintaining chain structure and internal memory and circulation [35]. It is widely used in traffic information prediction [36]. However, when the input sequence is long, RNN will have the problem of long-range dependence.

As a variant of RNN, Long Short-Term Memory (LSTM) effectively solves problems such as gradient dispersion of RNN and can make better use in long-distance time-series data [37]. LSTM neural network was first proposed in 1997 and is a special form of RNN. Compared with other neural networks, LSTM has better applicability in processing sequence data and identifying trends [38]. LSTM model has been successfully applied in time series data research in various fields, including traffic flow prediction in the field of road transportation [38], text speech recognition and machine translation in the field of text language [39], and protein structural sequence prediction in the field of medicine [40].

In this paper, we put up with a hybrid network named Driving Behavior Risk Prediction Neural Network (DBRPNN), Based on the LSTM model, and the rest parts are organized as follows: The third part is the description of the distracted driving behavior data, the fourth part is the DBRPNN structure, the fifth part is the results and discussion, the sixth part is the application and future implementation, and the last part is the conclusion.

## Data description

### Distracted driving behavior data

The information collected by multiple camera sensors and radar sensors of the vehicle can be used to obtain the distracted driving behavior data after image recognition and distance recognition. The main purpose of the collection of the distracted driving behavior data is to comprehensively record the information of the distracted conditions of the driver and the vehicle and to remind timely. The complete distracted driving behavior data usually needs to be determined by the time of the behavior, the latitude and longitude of the behavior, and the behavior code. The typical structure of the distracted driving behavior data (taking Shaanxi Province of China as an example) is shown in Table 1. This article does not list some irrelevant fields, such as vehicle registration, owner, speed, etc.

Due to a variety of possible situations, the distracted driving behavior data generates abnormal data that can affect the results. The following cases of data will be deleted in this paper to reduce interference.

(1) Data loss: it cannot reflect the specific situation of the distracted driving behavior.

(2) Data redundancy: multiple data records reflect the same distracted driving behavior.

(3) Data anomaly: data records violate normal travel rules. Including record data when the vehicle is not driving, the latitude and longitude are not within the normal range, etc.

The time record in the distracted driving behavior data is intermittent. To observe the prediction effects under different periods, this paper refers to some similar research practices [41, 42], combines the distracted driving behavior data according to vehicle ID or area ID, and summarizes it into four different time intervals: 30 minutes interval, 60 minutes interval, 90 minutes interval, and 120 minutes interval. When the time interval is less than 30 minutes, the scale of time units containing distracted driving behavior will be small. In the model validation section, we will also study the model performance differences at different time intervals.

### Attributes data

Right attributes are significant for describing factors related to distracted driving behaviors, which is conducive to the prediction of distracted driving behaviors [30]. Many studies have proved that the occurrence of distracted driving behavior is related to the external environment and the driver's mental state [6, 8]. In this paper, the time of a day is used to describe the factor of the external environment. Due to the difference in visibility between day and night, and the number of distracted driving behaviors in the morning is different from that in the afternoon, the number of distracted driving behaviors in the first half of the night is different

**Table 1. Distracted driving behavior data structure.**

| Field Name | Field Type | Data Example | Remarks Example |
|---|---|---|---|
| Vehicle Id | Int | 16254 | |
| Vehicle Trans Type | Int | 10 | '10'- Passenger vehicles, '30'- Dangerous goods transport vehicle |
| Time | Data | 2021-03-01 05:41:06 | |
| Code | Int | 10404 | |
| Longitude | Float | 107.203014 | |
| Latitude | Float | 34.369448 | |

from that in the second half of the night. Therefore, a day is divided into four time periods at an interval of 6 hours. The quantification of specific variables is shown in Table 2. Since it is difficult to obtain the mental state of drivers, the fatigue state of individuals is deeply affected by driving intensity [43], so this paper uses the daily driving time and daily mileage to measure this attribute. As the input of the neural network, the attribute of the time of a day is called DayID, and the other two attributes are summarized as a whole set, called DriveID. The sample data is shown in Table 3.

## Network structure of DBRPNN

### Definition, distracted driving behavior categories

The distracted driving behavior categories refer to the factors that threaten driving safety detected in the process of road transportation. There are eight distracted driving behavior codes used in this paper, as shown in Table 4. According to different behavior objects, distracted driving behaviors can be divided into two categories. Three codes of behaviors in category 103 are the distracted driving behavior shown by the vehicle, and five codes of behaviors in category 104 are the distracted driving behavior shown by the driver.

### Network establishment

Due to the different driving habits of drivers and types of vehicles, the number of distracted driving behaviors cannot be a good measure of the risk status of distracted driving behaviors. Therefore, this paper will predict the risk level. The neural network used to predict the risk level is a modular plug-in neural network. The architecture is shown in Fig 1. It consists of three modules, the Feature Processing Module (FPM), the Memory Module (MM), and the Prediction Module (PM). The FPM is a module responsible for classifying and standardizing features, and connecting them in series. The MM is based on LSTM to capture the time dependence of risk level changes. The PM is responsible for converting the output of the neural network into a risk level.

### Feature Processing Module (FPM)

**Hierarchical & normalization & concatenate.** The risk level is used to describe the degree of danger during driving. This paper uses the K-means algorithm to classify the risk levels. K-means is a clustering algorithm that determines the category of feature parameters based on the distance between each point in the data feature parameter set and the cluster center. This paper summarizes the historical distracted driving behavior data according to different time intervals and obtains the number of distracted driving behaviors $N_{i_t}^C$ for each vehicle in the period $i_t$, $i_t$ represents the t-th period divided by the i time interval. Use K-means to classify the number of distracted driving behaviors for each vehicle, set the k value to 3, and get the risk level $L_{i_t}^C$ of each vehicle in the period $i_t$, which is 0, 1, and 2 respectively. In addition, this article also divides the 100*100 grid according to the latitude and longitude range of the

**Table 2. Quantization of time.**

| The variable name | Quantitative range | Quantitative coding |
|:---:|:---:|:---:|
| The time of a day | [0:00 ~ 6:00) | 0 |
| | [6:00 ~ 12:00) | 1 |
| | [12:00 ~ 18:00) | 2 |
| | [18:00 ~ 00:00) | 3 |

**Table 3. The input data structure of DBRPNN.**

| Field Name | Field Type | Data Example |
|---|---|---|
| Risk Level | Int | 0 |
| The Time of a Day | Int | 3 |
| Driving Time | Float | 8.77 |
| Driving Mileage | Float | 318.2 |

distracted driving behavior point, and obtains the average number of vehicle distracted driving behaviors $N_{i_t}^A$ in each grid during the period. Use K-means to classify the average number of vehicle distracted driving behaviors in each area, set the k value to 3, and obtain the risk level $L_{i_t}^A$ of each area, which is 0, 1, and 2 respectively. Level 0 indicates that the vehicle or area is in a low-risk state at this time and no action is required. Level 1 indicates that the vehicle or area is in a medium-risk state and measures should be taken according to the situation. Level 2 indicates that the vehicle or area is in a high-risk state and immediate measures are required.

This paper uses Silhouette Coefficient to evaluate the clustering effect of K-means, and the results of different values of k are shown in Table 5.

The driving time of the day and the driving distance of the day in DriveID are the required factors, but their format is not suitable for direct input to the neural network, and must be normalized. This paper uses Z-score normalization to avoid extreme value changes in the network weight. Speed up the training process. DriveID and Day ID together constitute the attribute set $D_{i_t}$. The formula of Z-score:

$$\hat{x} = \frac{x - E(x)}{\sqrt[2]{Var(x)}} \tag{1}$$

Concatenate here is used to merge the risk level with DayID and Drive ID into the entire carrier before capturing time dependence.

## Memory Module (MM)

In the prediction of risk levels, the output of the network is not only related to the input at the current moment but also related to the output in the past period. RNN is a neural network with short-term memory capabilities. The neurons in RNN can not only receive information from other neurons but also receive their information, forming a cyclic network structure. LSTM is a variant of RNN, which can effectively solve the gradient dispersion problem of the simple recurrent neural network, and can better characterize time series data. Based on RNN,

**Table 4. Categories of distracted driving behavior.**

| Category | Code | Paraphrase |
|---|---|---|
| 103 | 10300 | About to hit the vehicle ahead while driving. |
| | 10301 | The vehicle deviates from the lane while driving. |
| | 10302 | Driving too close to the vehicle ahead. |
| 104 | 10400 | The driver is driving with physical fatigue. |
| | 10401 | The driver makes calls while driving. |
| | 10402 | The driver smokes while driving. |
| | 10403 | The driver closes his eyes while driving. |
| | 10404 | The driver yawns while driving. |

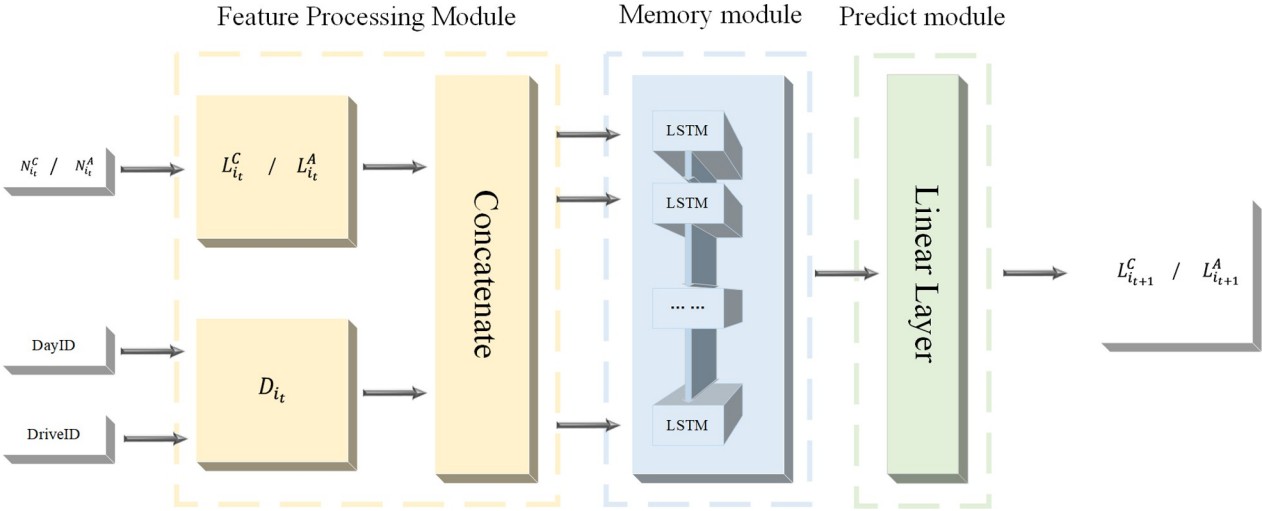

**Fig 1. Driving behavior risk prediction neural-network (DBRPNN) architecture.**

the improvement of LSTM mainly lies in two aspects, the introduction of a new internal state and the introduction of a gating mechanism.

LSTM introduces a new internal state $c_t \in R^n$ (n-dimensional column vector) specifically for linear cyclic information transfer, and at the same time outputs information nonlinearly to the external state $h_t \in R^n$ of the hidden layer. The internal state $c_t$ records the historical information up to the current moment, the calculation formula:

$$c_t = f_t \odot c_{t-1} + e_t \odot \tilde{c}_t \tag{2}$$

$$h_t = o_t \odot \tanh(c_t) \tag{3}$$

Where: $f_t \in [0,1]^n$ is the forgetting gate, which controls how much information should be forgotten in the internal state $c_{t-1}$ at the last moment; $e_t \in [0,1]^n$ is the input gate, which controls how much information the candidate state $\tilde{c}_t$ should keep at the current moment; $o_t \in [0,1]^n$ is the output gate, which controls how much information the internal state $c_t$ should output to the external state $h_t$ at the current moment, and the three gates control information transmission Path; $\odot$ is the product of vector elements; $c_{t-1}$ is the memory unit at the previous moment; $\tilde{c}_t \in R^n$ is the candidate state obtained by the nonlinear function, the calculation formula:

$$\tilde{c}_t = tanh(W_c x_t + U_c h_{t-1} + b_c) \tag{4}$$

Where: $W$, $U$, $b$ are the learnable network parameters.

LSTM introduces a gating mechanism to control the path of information transmission. The three gates are soft gates with values between (0,1), allowing information to pass through in a

**Table 5. The clustering effect of K-means.**

| Number of Clusters | Silhouette Score |
|---|---|
| 2 | 0.8193 |
| 3 | 0.8200 |
| 4 | 0.8033 |
| 5 | 0.8085 |

certain proportion. The calculation formula of the three gates:

$$f_t = \sigma(W_f x_t + U_f h_{t-1} + b_f) \tag{5}$$

$$e_t = \sigma(W_i x_t + U_i h_{t-1} + b_i) \tag{6}$$

$$o_t = \sigma(W_o x_t + U_o h_{t-1} + b_o) \tag{7}$$

Where: $\sigma$ is the Logistic function, and the output interval of the function is (0,1); $x_t$ is the input at the current moment, and $h_{t-1}$ is the external state at the last moment.

The recurrent unit structure of LSTM is shown in Fig 2. The calculation process is: firstly calculate the three gates and $\tilde{c}_t$ through $x_t$ and $h_{t-1}$, namely formula (4) to formula (7); then combine $f_t$ and $e_t$ to update $c_t$, namely formula (2); finally, combine $o_t$ passes information to $h_t$, which is formula (3).

## Prediction module

This module is responsible for converting the output of LSTM into a risk level $L^C_{i_{t+m}}$ (or $L^A_{i_{t+m}}$). Here a linear layer is used to convert the neuron output into three risk levels through Linear Mapping.

## Model training

This paper evaluates the performance of the network through Accuracy and Weighted-Precision. Accuracy is the percentage of samples with correct predictions to the total samples. Precision in the second classification is the percentage of the actual positive samples predicted to be positive. This paper is the three classification situation, so the Precision of each level needs to

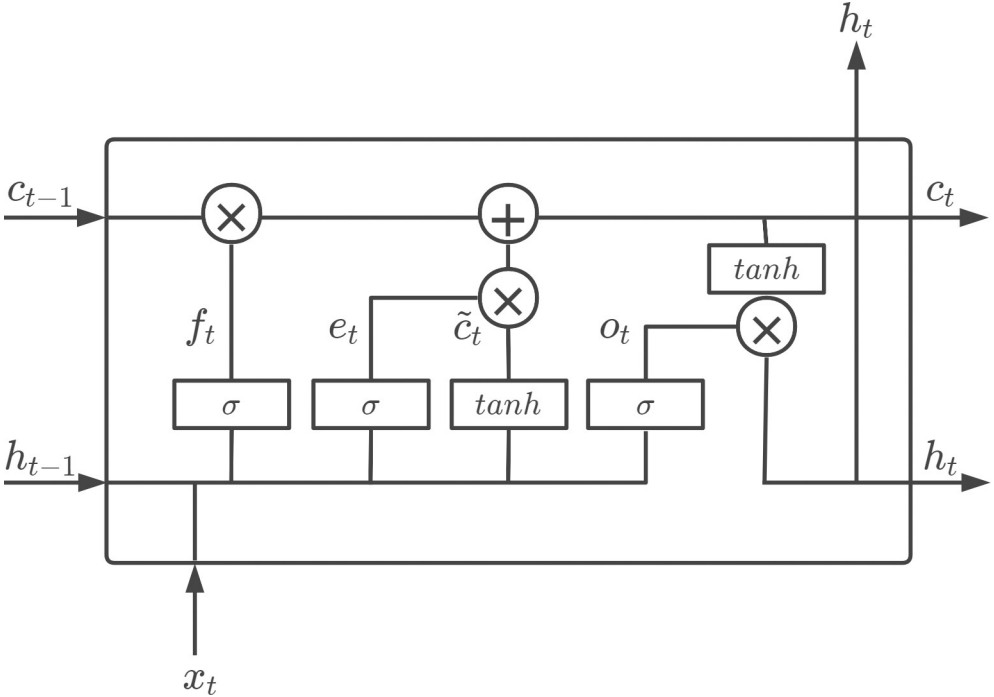

**Fig 2. LSTM network architecture.**

be calculated and weighted. The calculation formula of the two:

$$Accuracy = \frac{T}{T + F} \tag{8}$$

$$\text{Weighted Precision} = \frac{TP_0}{TP_0 + FP_0} * W_0 + \frac{TP_1}{TP_1 + FP_1} * W_1 + \frac{TP_2}{TP_2 + FP_2} * W_2 \tag{9}$$

Where: $T$ is the number of samples with correct grade prediction; $F$ is the number of samples with incorrect grade prediction; $TP_x$ ($x$ is 0, 1, 2) is the number of positive samples predicted to be positive samples for various types, $FP_x$ ($x$ is 0, 1, 2) is the number of various types of actual positive samples predicted as negative samples, and $W_x$ ($x$ is 0, 1, 2) is the proportion of each type to the total number of vehicles.

## Experiment establishment

### Experimental data description

This paper uses the distracted driving behavior data of 74 passenger transport vehicles and 26 dangerous goods transport vehicles in Shaanxi Province to conduct an empirical study. This data set is collected and managed by the Department of Transport Shaanxi Province and provided by Shaanxi Provincial Road Transport Development Center. Shaanxi Province is an inland province in northwestern China. There are 10 prefecture-level cities and 107 county-level administrative regions. Shaanxi Province is an important province connecting the northwestern region to other regions.

Since most of these 100 vehicles are transported in medium and long distances, the location where the distracted driving behavior occurred is mainly in Shaanxi Province and spread across many provinces around Shaanxi. The distracted driving behavior coordinate points are shown in Fig 3. There are eight codes of distracted driving behavior in this paper, and the frequency diagram of each code is shown in Fig 4. The data used in this paper covers 92 days (March, April, and May 2020), with an average of 5,705 pieces of data per day. We use the data of the first 72 days for training, the data of the next 10 days as the validation set, and the data of the last 10 days for testing. When we perform neural network training and model verification, we will delete abnormal data according to the situation mentioned in the description section. Fig 5 is a graph of the daily average number of distracted driving behaviors throughout the network.

### Experimental settings

DBRPNN is built with PyTorch 3.9, a well-known AI platform. In the training process, the loss function is Cross-Entropy, the optimizer is Adam whose learning rate is set as 0.01, and the batch size is 64. EPOCH is set to 100, and Early Stopping is used to control the number of iterations to prevent overfitting. Training ends when the loss function no longer improves or EPOCH reaches 100. The training computer equipped with a Graphics 630 and one Intel Core i5 CPU. The operation system is Windows 10.

### Performance evaluation

Using DBRPNN to predict the risk level of vehicles and areas, some of the results are shown in Fig 6. It can be seen from the figure that DBRPNN's prediction results for different objects have the same trend as the real data.

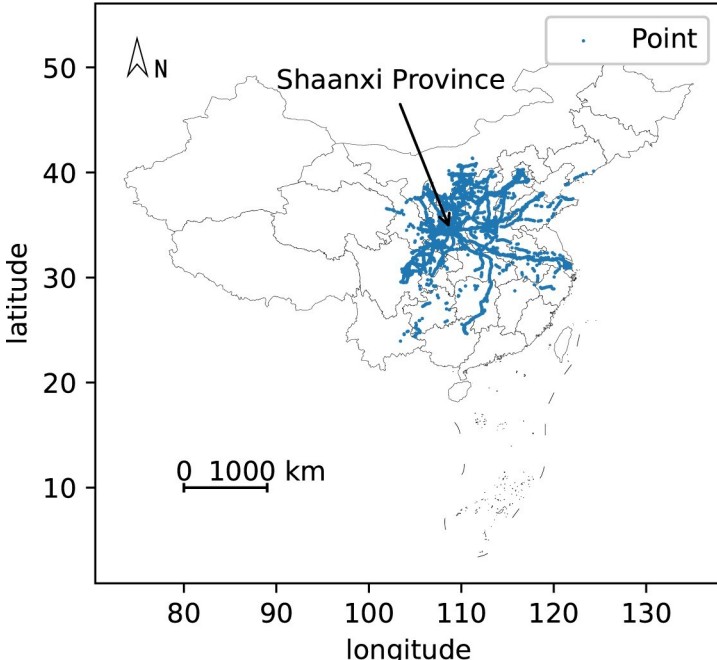

**Fig 3. Distracted driving behavior coordinate point.** The coordinates of vehicles at the time of distracted driving behavior can be seen in the figure, which is mainly distributed in Shaanxi Province and some surrounding provinces.

To make the results more convincing, the average of the predicted results of 10 vehicles is selected to evaluate the network performance, as shown in Table 6.

- **CART**: Employ CART model to predict the level of risk within a specific time interval.

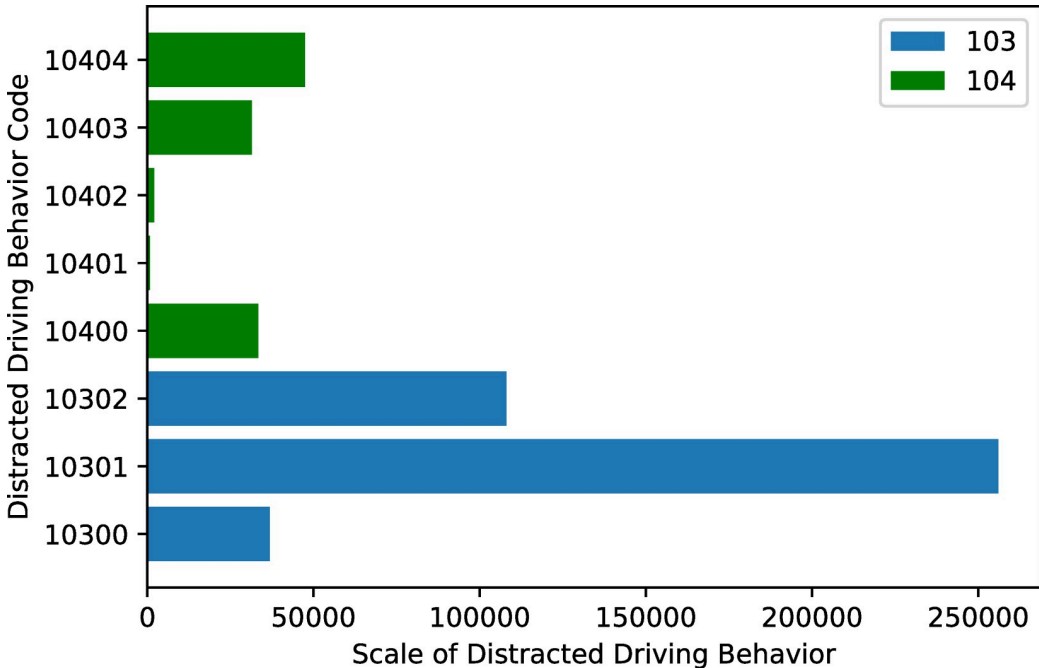

**Fig 4. Scale of each distracted driving behavior code.**

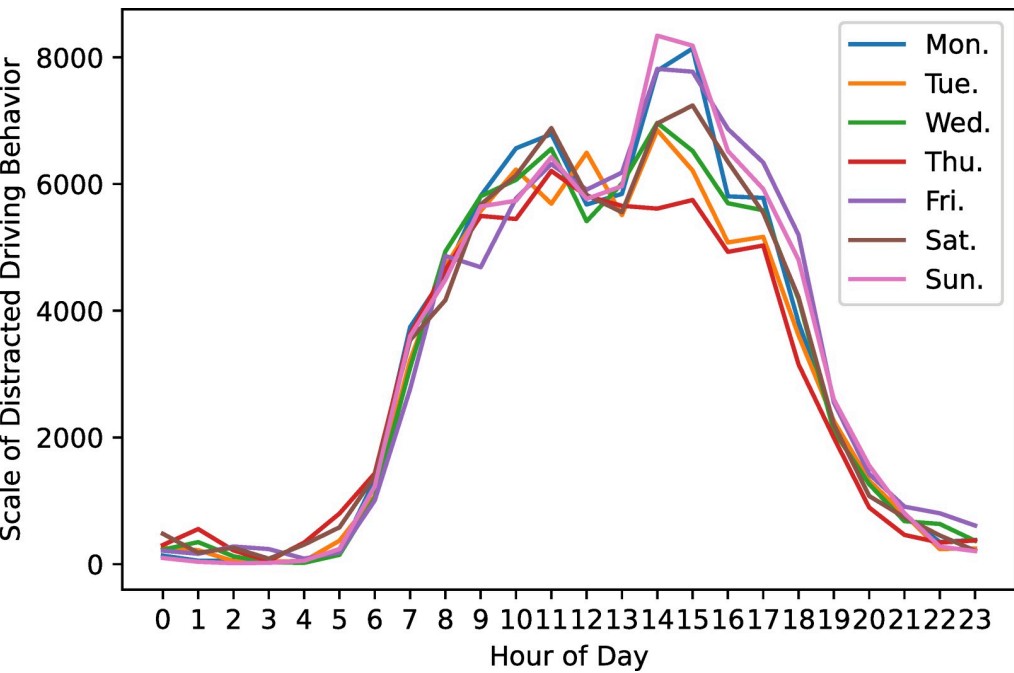

**Fig 5. Daily average number of distracted driving behaviors.**

- **SVM**: Employ SVM model to predict the level of risk within a specific time interval.

- **RNN**: Take advantage of RNN to predict the level of risk within a specific time interval.

- **1 Layer LSTM and 2 Layer-LSTM**: Use one and two layers LSTM networks to predict the level of risk. The result shows that a 2-layers LSTM is better than one-layer.

- **Bi-LSTM**: Use bidirectional LSTM (Bi-LSTM) network without the FPM. The prediction result of Bi-LSTM is better than that of ordinary LSTM.

From the results, it can be found that the DBRPNN network can effectively predict the risk level. And with the help of the FPM, Accuracy is increased by 5.42% compared with only use the Bi-LSTM.

This paper also trains DBRPNN through four different time interval data, and tests each trained DBRPNN. The results are shown in Table 7. The 30 minutes prediction has the highest Accuracy of 0.9146, which is encouraging. In addition, DBRPNN also has stable and good performance in different time intervals.

Under normal circumstances, different distracted driving behaviors will lead to large deviations in the risk level prediction. To measure the performance of DBRPNN in different situations, we divided the codes of distracted driving behaviors into two categories: the distracted driving behavior shown by the vehicle (103) and the distracted driving behavior shown by the driver (104). The comparative performance is shown in Table 8.

From the result, the prediction Accuracy of Category 104 in the area is relatively low, because the driver's behavior is more random and the number of vehicles passing through the area fluctuates greatly, so it is difficult to predict. But in addition to this, DBRPNN has very high prediction results (accuracy rate greater than 0.9).

During the prediction process, it was found that there were vehicles with all prediction results of level 0. The study found that this type of vehicle rarely had distracted driving behaviors, and the actual risk level was level 0 during the test time. The distracted driving behavior

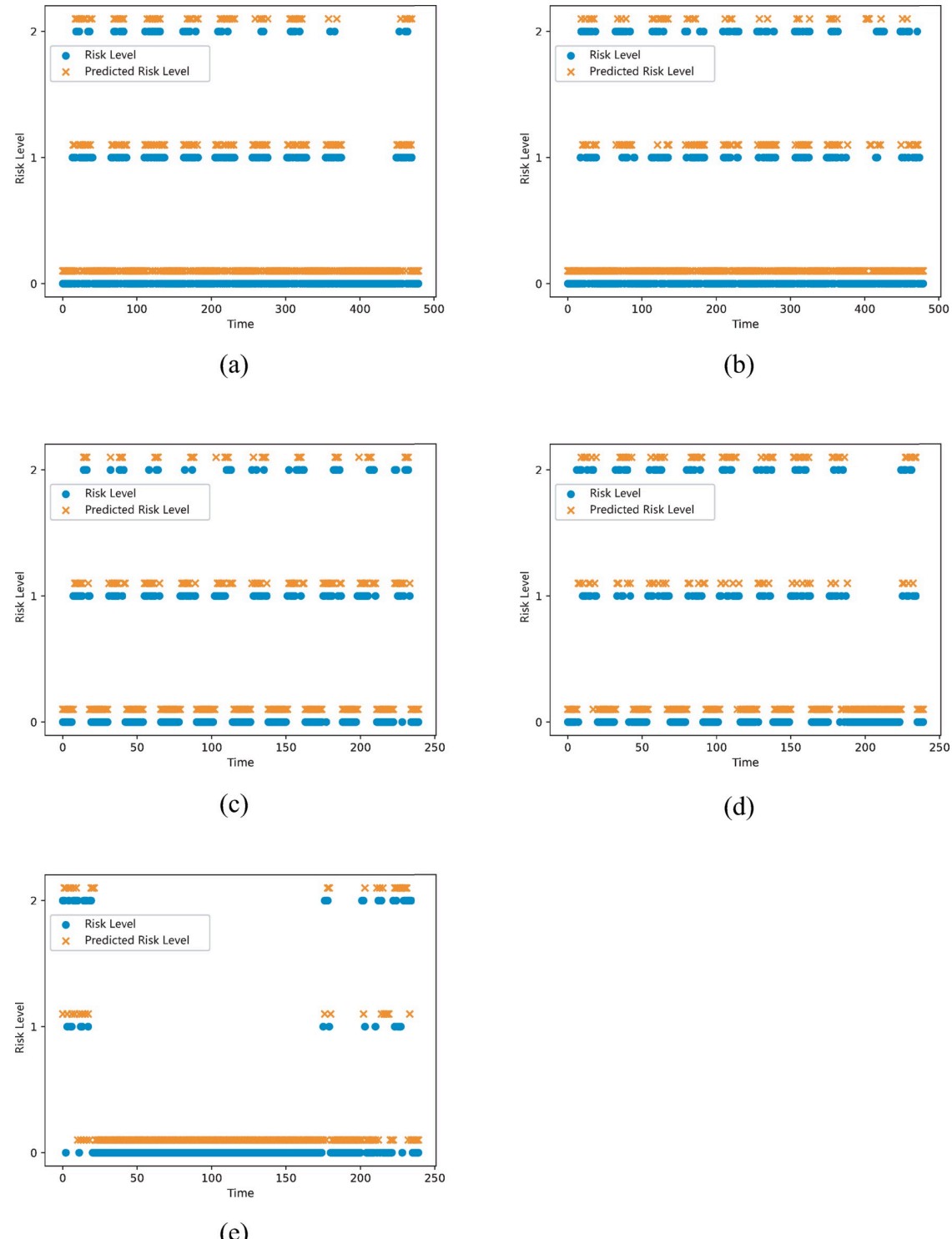

**Fig 6. The prediction result of the risk level.** (a) Vehicle ID 6096. (b) Vehicle ID 21973. (c) Area ID (53,24). (d) Area ID (53,23). (e) Category 104 of Vehicle ID 20635.

of this type of vehicle is difficult to measure by predicting the results, and every time a distracted driving behavior occurs, timely measures should be taken.

**Table 6. Network performance comparison.**

|  | Accuracy | Weighted-Recall |
|---|---|---|
| CART | 0.7712 | 0.7724 |
| SVM | 0.7866 | 0.8135 |
| RNN | 0.7979 | 0.8137 |
| 1 Layer-LSTM | 0.8214 | 0.8116 |
| 2 Layer-LSTM | 0.8583 | 0.8479 |
| Bi-LSTM | 0.8604 | 0.8665 |
| DBRPNN | 0.9146 | 0.9156 |

## Application and implementation

In this study, we use the warning data of distracted driving behavior and the corresponding attribute data to construct a DBRPNN to realize the risk level prediction of different time granularities. Judging from the application and implementation of this method, this kind of forecasting work helps to improve the safety of road driving. Due to the bad driving habits of some drivers, multiple distracted driving behaviors will often occur during the driving process. If the driver is reminded every time a distracted driving behavior occurs, it will consume a lot of manpower and the effect is not good, but it should not be ignored. DBRPNN can accurately predict the risk level of each period of the vehicle. Reminders are given in advance in the period with higher risk level, and the driver is contacted in time when the vehicle distracted driving behaviors during the high-risk period, which can effectively prevent accidents and improve the active safety of transportation. Distracted driving behaviors that occur in vehicles or areas are closely related to accidents. Therefore, high-precision, variable time-granularity risk prediction has an irrefutable impact on road accident prevention and has a positive impact on the active safety of road transportation. The method proposed in this paper can predict the risk of driving behavior based on the distracted driving behavior data within a certain precision and accuracy range. Compared with some methods and experimental analysis, it has more practical and effective effects and can be a single vehicle or area that provides forecast results. In most cases, vehicles are managed by local transportation departments. In a local transportation department's network, a relatively complete distracted driving behavior data collection system has been formed. Therefore, the method proposed in this paper can be transplanted to local active safety early warning systems to provide predictions on transportation safety. It will help local transportation departments to improve transportation safety, including reducing human resource investment and realizing active safety with higher efficiency. It can also predict the risk level of the area and inform in advance the vehicles that will pass the area during the high-risk period, thereby reducing the traffic risk in the area. In addition, we believe that another noteworthy problem of this work is to provide a network prediction solution based on spatiotemporal data. This method is also suitable for predictions with the same type of data, such as accident prediction. It also helps to build other types of prediction networks based on spatiotemporal data sets and attribute data.

**Table 7. Prediction performance for different time intervals.**

| Time interval | Accuracy | Weighted-Recall |
|---|---|---|
| 30 min | 0.9146 | 0.9156 |
| 60 min | 0.8854 | 0.8843 |
| 90 min | 0.8833 | 0.8756 |
| 120 min | 0.8542 | 0.8810 |

**Table 8. Prediction performance for different categories of distracted driving behaviors.**

| Predicted Object | Category | Accuracy | Weighted-Recall |
|---|---|---|---|
| Vehicle | 103 | 0.9229 | 0.9258 |
| | 104 | 0.9083 | 0.9105 |
| Area | 103 | 0.9042 | 0.9078 |
| | 104 | 0.8875 | 0.8975 |

## Conclusion

This paper proposes an LSTM-based Driving Behavior Risk Prediction Neural Network (DBRPNN). Improve the accuracy of prediction by combining time attributes and vehicle attributes. Using the provincial proportional data set to train and test DBRPNN, the results show that DBRPNN has a stable and encouraging Accuracy, which can predict risk be based on different time intervals (30 minutes, 60 minutes, 90 minutes, 120 minutes) and different categories (Category 103, Category 104) with relatively high Accuracy. In addition, the implementation of DBRPNN has broad prospects, and artificial intelligence technology has once again demonstrated its power in the field of transportation. Researchers will continue to improve the Accuracy of DBRPNN and use other advanced neural networks to further study driving behavior risk prediction.

## Author Contributions

**Conceptualization:** Xin Fu, Hongwei Meng.

**Data curation:** Xin Fu.

**Formal analysis:** Hongwei Meng.

**Resources:** Jianwei Wang.

**Software:** Jianwei Wang.

**Validation:** Hao Yang.

**Writing – original draft:** Xue Wang.

**Writing – review & editing:** Hao Yang.

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
