## [Decision Letter · Decision Letter 0]

17 Nov 2021

PONE-D-21-34056A Hybrid Neural Network for Driving Behavior Risk Prediction Based on Distracted Driving Behavior DataPLOS ONE

Dear Dr. Meng,

Thank you for submitting your manuscript to PLOS ONE. After careful consideration, we feel that it has merit but does not fully meet PLOS ONE’s publication criteria as it currently stands. Therefore, we invite you to submit a revised version of the manuscript that addresses the points raised during the review process.

We look forward to receiving your revised manuscript.

Kind regards,

Feng Chen

Academic Editor

PLOS ONE

Journal Requirements:

"The research is supported by Key R&D Project of the Ministry of Science and Technology of China (Granted No. 2020YFC1512004)."

"There are no financial conflicts of interest to disclose."

3. We note that Figure 3 in your submission contain [map/satellite] images which may be copyrighted. All PLOS content is published under the Creative Commons Attribution License (CC BY 4.0), which means that the manuscript, images, and Supporting Information files will be freely available online, and any third party is permitted to access, download, copy, distribute, and use these materials in any way, even commercially, with proper attribution. For these reasons, we cannot publish previously copyrighted maps or satellite images created using proprietary data, such as Google software (Google Maps, Street View, and Earth). For more information, see our copyright guidelines: http://journals.plos.org/plosone/s/licenses-and-copyright.

a. You may seek permission from the original copyright holder of Figure 3 to publish the content specifically under the CC BY 4.0 license.  

Reviewers' comments:

Reviewer's Responses to Questions

**Comments to the Author**

1. Is the manuscript technically sound, and do the data support the conclusions?

Reviewer #1: Partly

Reviewer #2: Yes

2. Has the statistical analysis been performed appropriately and rigorously? 

Reviewer #1: Yes

Reviewer #2: Yes

3. Have the authors made all data underlying the findings in their manuscript fully available?

Reviewer #1: Yes

Reviewer #2: Yes

4. Is the manuscript presented in an intelligible fashion and written in standard English?

Reviewer #1: Yes

Reviewer #2: Yes

5. Review Comments to the Author

Reviewer #1: This study proposes a driving risk prediction algorithm based on a popular Recurrent Neural Network architecture, LSTM. The proposed algorithm is demonstrated by comparing with several selected baseline models. The research idea is excellent, but some concerns need to be addressed for the potential publication. The major comments are listed below.

1, In the study, driving risk is categorized into 0, 1, and 2. Please clearly introduce the definition of driving risk level in the manuscript.

2, Please introduce the input data of DBRPNN, and provide a sample data as the illustration.

3, To my understanding, the purpose of predicting driving behavior risk is to provide timely warning information for avoiding potential safety issues. Thus, the predicted information is valid for a short-term period. However, in the study, the authors present the prediction for 30 mins, 60 mins, 90 mins, and 120 mins. Please explain why the authors select these four time-intervals for performance demonstration.

4, Figure 3, “Shaanxi Province” should be a typo. Figure 1 should be replaced by a high-resolution figure to improve the readability.

5, The manuscript contains considerable language issues. Please conduct a proofreading and polish the manuscript accordingly.

Reviewer #2: This manuscript presents an interesting and meaningful piece of research,

a method to predict distracted driving behavior is proposed. I think with the increasingly rich data of the Internet of Things, more risk collection data can provide better early warning support. The paper is well written and I have the following comments:

(1) The literature review is obviously not comprehensive enough. In terms of risk prediction, behavioral prediction and flow prediction have made abundant research progress, but they are not well reflected in the review section.

(2) Why are risk levels divided into three categories in clustering? The paper is not clear on this point.

(3) If the classification criteria of risk level are changed, will the accuracy of the results change?

(4) "The driving behavior of this type of vehicle is difficult to measure by predicting The results, And every time a driving behavior occurs, timely measures should be taken." Why is that?

(5) The reference in this study needs further expansion, several papers fousing on prediciton using deep learning modles may be useful for their further studies.

[1]Spatiotemporal gated graph attention network for urban traffic flow prediction based on license plate recognition data”. Computer-Aided Civil and Infrastructure Engineering, 2021, 1-21. DOI: 10.1111/mice.12688.

[2]Multi- community passenger demand prediction at region level based on spatio-temporal graph convolutional network”, Transportation Research Part C: Emerging Technologies, Vol.124, 102951, 2021, 1-18.

I hope these questions can contribute to the further improvement of this manuscript.

6. PLOS authors have the option to publish the peer review history of their article (what does this mean?). If published, this will include your full peer review and any attached files.

Reviewer #1: No

Reviewer #2: No

---

## [Author Response · Author response to Decision Letter 0]

24 Dec 2021

Reply Letter

Dear Editor,

We quite appreciate your favorite consideration and the reviewers’ insightful comments concerning our manuscript entitled “A Hybrid Neural Network for Driving Behavior Risk Prediction Based on Distracted Driving Behavior Data”. Those comments are very valuable and helpful for improving the quality and readability of our paper, as well as the important guiding significance to our future researches. The modifications are marked in blue, red colors according to each reviewer, respectively in the revised manuscript and this reply letter. Moreover, detailed point-by-point responses to all of the comments from reviewers are provided. Here, a brief overview and summary of the revisions are shown as follows.

Based on the comments from the Reviewers and Editor, the manuscript has been well revised.

We hope this revised manuscript has addressed your concerns, and look forward to hearing from you.

Sincerely,

Hongwei Meng

Response to Editor’s Comments

Academic Editor’s Comments:

Thank you for submitting your manuscript to PLOS ONE. After careful consideration, we feel that it has merit but does not fully meet PLOS ONE’s publication criteria as it currently stands. Therefore, we invite you to submit a revised version of the manuscript that addresses the points raised during the review process.

Response:

Thank you very much for the comments and the chance to improve our manuscript. We have checked the manuscript and corrected the content, the grammar, and the figures. The modifications corresponding to Reviewer 1’s comments are marked in blue, Reviewer 2's are marked in red in the revised manuscript and this reply letter. Detailed point-by-point responses to your comments are provided below.

1, Please ensure that your manuscript meets PLOS ONE's style requirements, including those for file naming.

Response:

We have made our manuscripts meet the style requirements of PLOS ONE, including file naming requirements.

2, Thank you for stating the following in the Acknowledgments Section of your manuscript:

"The research is supported by Key R&D Project of the Ministry of Science and Technology of China (Granted No. 2020YFC1512004)."

Please remove any funding-related text from the manuscript and let us know how you would like to update your Funding Statement. Currently, your Funding Statement reads as follows: "There are no financial conflicts of interest to disclose."

Response:

Thank you very much. We have removed the funding information from the manuscript and have included our amendment statement in the cover letter.

(3) We note that Figure 3 in your submission contain [map/satellite] images which may be copyrighted.

Response:

The map image in Figure 3 is generated from OSM and is open source.

OpenStreetMap: https://www.openstreetmap.org/

 

Response to Reviewer #1’s Comments

Comments to the Author:

This study proposes a driving risk prediction algorithm based on a popular Recurrent Neural Network architecture, LSTM. The proposed algorithm is demonstrated by comparing with several selected baseline models. The research idea is excellent, but some concerns need to be addressed for the potential publication. The major comments are listed below.

1, In the study, driving risk is categorized into 0, 1, and 2. Please clearly introduce the definition of driving risk level in the manuscript.

2, Please introduce the input data of DBRPNN, and provide a sample data as the illustration.

3, To my understanding, the purpose of predicting driving behavior risk is to provide timely warning information for avoiding potential safety issues. Thus, the predicted information is valid for a short-term period. However, in the study, the authors present the prediction for 30 mins, 60 mins, 90 mins, and 120 mins. Please explain why the authors select these four time-intervals for performance demonstration.

4, Figure 3, “Shaanxi Province” should be a typo. Figure 1 should be replaced by a high-resolution figure to improve the readability.

5, The manuscript contains considerable language issues. Please conduct a proofreading and polish the manuscript accordingly.

Response:

We feel great thanks for your professional review work on our article. As you are concerned, there are several problems that need to be addressed. According to your comments, the manuscript has been revised. The modifications corresponding to your comments are marked in blue. Detailed point-by-point responses to your comments are provided below. 

1, In the study, driving risk is categorized into 0, 1, and 2. Please clearly introduce the definition of driving risk level in the manuscript.

Response:

Thank you again for your positive comments and valuable suggestions to improve the quality of our manuscript.

In our study, driving risk is divided into 0, 1, and 2 by K-means. Level 0 indicates that the vehicle or area is in a low-risk state at this time and no action is required. Level 1 indicates that the vehicle or area is in a medium-risk state and measures should be taken according to the situation. Level 2 indicates that the vehicle or area is in a high-risk state and immediate measures are required. 

We have added an introduction to the definition of driving risk level in the revised manuscript, shown as follows.

Level 0 indicates that the vehicle or area is in a low-risk state at this time and no action is required. Level 1 indicates that the vehicle or area is in a medium-risk state and measures should be taken according to the situation. Level 2 indicates that the vehicle or area is in a high-risk state and immediate measures are required.

2, Please introduce the input data of DBRPNN, and provide a sample data as the illustration.

Response:

We are very grateful for your comments. It is necessary to display the input of DBRPNN, which will make our paper clearer.

The input data of DBRPNN includes risk level, the time of a day, driving time, and driving mileage. The risk level is divided into 0, 1, and 2. The time of the day is divided into four time periods: 0: 00 to 6:00, 6:00 to 12:00, 12:00 to 18:00, 18:00 to 24:00, respectively, using 0, 1, 2, and 3 to indicate. The driving time is the length of time the driver drives the vehicle. The driving mileage is the distance the driver drives the vehicle. Driving time and driving mileage are used to measure the fatigue state of the driver. 

We provide sample data as an illustration in the revised manuscript, as shown below.

The sample data is shown in Table 3.

Table 3. The Input Data Structure of DBRPNN.

Field Name Field Type Data Example

Risk Level Int 0

The Time of a Day Int 3

Driving Time Float 8.77

Driving Mileage Float 318.2

3, To my understanding, the purpose of predicting driving behavior risk is to provide timely warning information for avoiding potential safety issues. Thus, the predicted information is valid for a short-term period. However, in the study, the authors present the prediction for 30 mins, 60 mins, 90 mins, and 120 mins. Please explain why the authors select these four time-intervals for performance demonstration.

Response:

Thank you for your comments. We think this is an excellent suggestion. 

Some similar research practices summarize the data into four different time intervals: 15 minutes interval, 30 minutes interval, 45 minutes interval, and 60 minutes interval. When we divide our data into 15 minutes interval, the number of periods during which distracted driving behavior occurs is less than 30% of the total number of periods. The data of the five vehicles with the largest number of distracted driving behaviors are shown in the table below. When there are too few non-zero data, predictability is low, so we choose a time interval of 30 minutes and above.

Vehicle ID Number of Periods During Which Distracted Driving Behavior Occurred Total Number of Periods

16759 2513 8640

16924 2509 

33028 2412 

20635 2339 

33024 2281 

We have revised the corresponding content and supplemented the relevant references, shown as follows.

To observe the prediction effects under different periods, this paper refers to some similar research practices [41, 42], combines the distracted driving behavior data according to vehicle ID or area ID, and summarizes it into four different time intervals: 30 minutes interval, 60 minutes interval, 90 minutes interval, and 120 minutes interval. When the time interval is less than 30 minutes, the scale of time units containing distracted driving behavior will be small.

42. Ma X, Yu H, Wang Y, et al. Large-Scale Transportation Network Congestion Evolution Prediction Using Deep Learning Theory. Plos One, 2015, 10.

4, Figure 3, “Shaanxi Province” should be a typo. Figure 1 should be replaced by a high-resolution figure to improve the readability.

Thanks for your careful checks. The spelling of "aa" is indeed confusing. According to the English naming standard of Chinese cities, the English name of this province is "Shaanxi Province".

In the revised manuscript, Figure 1 has been replaced by a high-resolution figure as follows.

Fig 1. Driving Behavior Risk Prediction Neural-Network (DBRPNN) Architecture.

5, The manuscript contains considerable language issues. Please conduct a proofreading and polish the manuscript accordingly.

We sincerely thank you for careful reading. We have tried our best to proofread and polish the revised manuscript. And here we did not list the changes but marked them in blue in the revised paper.

 

Response to Reviewer #2’s Comments

Comments to the Author:

This manuscript presents an interesting and meaningful piece of research, a method to predict distracted driving behavior is proposed. I think with the increasingly rich data of the Internet of Things, more risk collection data can provide better early warning support. The paper is well written and I have the following comments:

(1) The literature review is obviously not comprehensive enough. In terms of risk prediction, behavioral prediction and flow prediction have made abundant research progress, but they are not well reflected in the review section.

(2) Why are risk levels divided into three categories in clustering? The paper is not clear on this point.

(3) If the classification criteria of risk level are changed, will the accuracy of the results change?

(4) "The driving behavior of this type of vehicle is difficult to measure by predicting The results, And every time a driving behavior occurs, timely measures should be taken." Why is that?

(5) The reference in this study needs further expansion, several papers fousing on prediciton using deep learning modles may be useful for their further studies.

[1]Spatiotemporal gated graph attention network for urban traffic flow prediction based on license plate recognition data”. Computer-Aided Civil and Infrastructure Engineering, 2021, 1-21. DOI: 10.1111/mice.12688.

[2]Multi- community passenger demand prediction at region level based on spatio-temporal graph convolutional network”, Transportation Research Part C: Emerging Technologies, Vol.124, 102951, 2021, 1-18.

Response:

Thank you again for your positive comments and valuable suggestions to improve the quality of our manuscript. According to your comments, the manuscript has been revised. The modifications corresponding to your comments are marked in red. Detailed point-by-point responses to your comments are provided below.

(1) The literature review is obviously not comprehensive enough. In terms of risk prediction, behavioral prediction and flow prediction have made abundant research progress, but they are not well reflected in the review section.

Response:

Thank you for your nice comments on our article. This makes our literature review more comprehensive.

In Related Work, we introduce distracted driving behavior, methods for predicting distracted driving behavior and traffic accidents, and related neural networks. We have supplemented the content of relevant behavioral prediction and flow prediction in relevant work, shown as follows.

In terms of risk prediction, behavioral prediction and flow prediction have made abundant research progress. Tang proposed a forecasting framework named the spatiotemporal gated graph attention network to predict the urban traffic flow based on license plate recognition data [19]. In addition, Pu uses historical data to predict road surface friction [20, 21]. Tang used a geographically weighted Poisson quantile regression model to study the spatial heterogeneity and estimated the spatial impact on crash frequency [22].

We have supplemented the relevant references, shown as follows.

19. Tang J, Zeng J. Spatiotemporal gated graph attention network for urban traffic flow prediction based on license plate recognition data. Computer-Aided Civil and Infrastructure Engineering, 2021(7).

20. Pu Z, Liu C, Shi X, et al. Road surface friction prediction using long short-term memory neural network based on historical data. Journal of Intelligent Transportation Systems, 2020(1):1-12.

21. Pu Z, Cui Z, Wang S, et al. Time-aware gated recurrent unit networks for forecasting road surface friction using historical data with missing values. IET Intelligent Transport Systems, 2020, 14(4):213-219.

22. Tang J, Gao F, Liu F, et al. Spatial heterogeneity analysis of macro-level crashes using geographically weighted Poisson quantile regression. Accident Analysis & Prevention, 2020, 148:105833.

29. Tang J, Liang J, Liu F, et al. Multi-community passenger demand prediction at region level based on spatio-temporal graph convolutional network. Transportation Research Part C Emerging Technologies, 2021, 124(10):102951.

(2) Why are risk levels divided into three categories in clustering? The paper is not clear on this point.

Response:

We think this is an excellent suggestion. Thank you for pointing out our omissions.

We use Silhouette Coefficient to evaluate the effect of K-means clustering. The results are shown in table 5. It can be seen from the table that the results are best when the number of clusters is three.

Table 5. The Clustering Effect of K-means.

Number of Clusters Silhouette Score

2 0.8193

3 0.8200

4 0.8033

5 0.8085

We have supplemented relevant content in the revised manuscript, shown as follows.

This paper uses Silhouette Coefficient to evaluate the clustering effect of K-means, and the results of different values of k are shown in Table 5.

Table 5. The Clustering Effect of K-means.

Number of Clusters Silhouette Score

2 0.8193

3 0.8200

4 0.8033

5 0.8085

(3) If the classification criteria of risk level are changed, will the accuracy of the results change?

Response:

Thank you for your nice comments. If the classification criteria of risk level are changed, the accuracy of the results will change. The results are shown in the table below.

Number of Risk Levels Accuracy Weighted-Recall

2 0.9167 0.9169

3 0.9146 0.9156

4 0.8458 0.8355

5 0.7562 0.7566

(4) "The driving behavior of this type of vehicle is difficult to measure by predicting The results, And every time a driving behavior occurs, timely measures should be taken." Why is that?

Response:

Thank you for your comment, and our reply is as follows.

Distracted driving behavior of this type of vehicle rarely occurs. The prediction results of distracted driving behavior of some of these vehicles are shown in Fig 7. The distracted driving behavior of this type of vehicle is difficult to predict. So we need to remind them in time when they are distracted driving.

(a) (b)

Fig 7. The Prediction Result of the Risk Level. (a) Vehicle ID 16975. (b) Vehicle ID 16822.

(5) The reference in this study needs further expansion, several papers fousing on prediciton using deep learning modles may be useful for their further studies.

[1]Spatiotemporal gated graph attention network for urban traffic flow prediction based on license plate recognition data”. Computer-Aided Civil and Infrastructure Engineering, 2021, 1-21. DOI: 10.1111/mice.12688.

[2]Multi- community passenger demand prediction at region level based on spatio-temporal graph convolutional network”, Transportation Research Part C: Emerging Technologies, Vol.124, 102951, 2021, 1-18.

Response:

We sincerely appreciate the valuable comments. These papers are very helpful for our current and further research. 

We have checked the literature carefully and added more references in the Related Work part in the revised manuscript, shown as follows.

19. Tang J, Zeng J. Spatiotemporal gated graph attention network for urban traffic flow prediction based on license plate recognition data. Computer-Aided Civil and Infrastructure Engineering, 2021(7).

20. Pu Z, Liu C, Shi X, et al. Road surface friction prediction using long short-term memory neural network based on historical data. Journal of Intelligent Transportation Systems, 2020(1):1-12.

21. Pu Z, Cui Z, Wang S, et al. Time-aware gated recurrent unit networks for forecasting road surface friction using historical data with missing values. IET Intelligent Transport Systems, 2020, 14(4):213-219.

22. Tang J, Gao F, Liu F, et al. Spatial heterogeneity analysis of macro-level crashes using geographically weighted Poisson quantile regression. Accident Analysis & Prevention, 2020, 148:105833.

29. Tang J, Liang J, Liu F, et al. Multi-community passenger demand prediction at region level based on spatio-temporal graph convolutional network. Transportation Research Part C Emerging Technologies, 2021, 124(10):102951.

---

## [Decision Letter · Decision Letter 1]

11 Jan 2022

A Hybrid Neural Network for Driving Behavior Risk Prediction Based on Distracted Driving Behavior Data

PONE-D-21-34056R1

Dear Dr. Meng,

We’re pleased to inform you that your manuscript has been judged scientifically suitable for publication and will be formally accepted for publication once it meets all outstanding technical requirements.

Kind regards,

Feng Chen

Academic Editor

PLOS ONE

Additional Editor Comments (optional):

Reviewers' comments:

Reviewer's Responses to Questions

**Comments to the Author**

1. If the authors have adequately addressed your comments raised in a previous round of review and you feel that this manuscript is now acceptable for publication, you may indicate that here to bypass the “Comments to the Author” section, enter your conflict of interest statement in the “Confidential to Editor” section, and submit your "Accept" recommendation.

Reviewer #1: All comments have been addressed

Reviewer #2: All comments have been addressed

2. Is the manuscript technically sound, and do the data support the conclusions?

Reviewer #1: Yes

Reviewer #2: Yes

3. Has the statistical analysis been performed appropriately and rigorously? 

Reviewer #1: Yes

Reviewer #2: Yes

4. Have the authors made all data underlying the findings in their manuscript fully available?

Reviewer #1: No

Reviewer #2: Yes

5. Is the manuscript presented in an intelligible fashion and written in standard English?

Reviewer #1: Yes

Reviewer #2: Yes

6. Review Comments to the Author

Reviewer #1: (No Response)

Reviewer #2: In this revision, authors responded all my concerning comments, and I think the quality of the paper has improved largely and it can be accepted for its current condition.

7. PLOS authors have the option to publish the peer review history of their article (what does this mean?). If published, this will include your full peer review and any attached files.

Reviewer #1: No

Reviewer #2: No

---

## [Editor Report · Acceptance letter]

14 Jan 2022

PONE-D-21-34056R1 

A Hybrid Neural Network for Driving Behavior Risk Prediction Based on Distracted Driving Behavior Data 

Dear Dr. Meng:

I'm pleased to inform you that your manuscript has been deemed suitable for publication in PLOS ONE. Congratulations! Your manuscript is now with our production department. 

Kind regards, 

on behalf of

Dr. Feng Chen 

Academic Editor

PLOS ONE